# Wnt/PKC Signaling Inhibits Sensory Hair Cell Formation in the Developing Mammalian Cochlea

**DOI:** 10.3390/cells14120888

**Published:** 2025-06-12

**Authors:** Joanna F. Mulvaney, Erynn M. Layman, Farhana Feroze-Merzoug, Julia M. Abitbol, Jennifer M. Jones, Dara O’Connor, Florence Naillat, Seppo Vainio, Jeffrey S. Rubin, Matthew W. Kelley, Alain Dabdoub

**Affiliations:** 1Sunnybrook Research Institute, Sunnybrook Health Sciences Centre, 2075 Bayview Ave, Toronto, ON M4N 3M5, Canada; jfmulvaney@fastmail.com (J.F.M.); julia.abitbol@sri.utoronto.ca (J.M.A.); spockrocket@gmail.com (D.O.); 2Laboratory of Cochlear Development, NIDCD, National Institutes of Health, Bethesda, MD 20892, USA; erynnmckenzie@hotmail.com (E.M.L.); kelleymt@nidcd.nih.gov (M.W.K.); 3Laboratory of Cellular and Molecular Biology, NCI, National Institutes of Health, Bethesda, MD 20892, USA; merzoug_farhana_feroze@lilly.com (F.F.-M.); rubinj@mail.nih.gov (J.S.R.); 4Department of Biology, Saint Mary’s College, Notre Dame, IN 46556, USA; jrowsell@saintmarys.edu; 5Department of Biochemistry and Molecular Medicine, Biocenter and Infotech Oulu, University of Oulu, FIN-90014 Oulu, Finland; lorence.naillat@oulu.fi (F.N.); seppo.vainio@oulu.fi (S.V.); 6Department of Otolaryngology—Head and Neck Surgery, University of Toronto, Toronto, ON M5S 1A1, Canada; 7Department of Laboratory Medicine and Pathobiology, University of Toronto, Toronto, ON M5S 1A1, Canada

**Keywords:** mouse, hearing, cochlea, genes, hair cell

## Abstract

The establishment of cell fate and boundaries between cell types is an essential step in development and organogenesis. In the mammalian cochlea, a distinct boundary exists between a medial region of non-sensory cells and a lateral region of sensory cells. We report that Wnt4 and sFRP2 act in combination to modulate the sensory cell differentiation of the organ of Corti. The hair cell inhibitory effects of Wnt4 in the inner ear are mediated through the activation of the non-canonical Wnt/Calcium/PKC pathway. We show that Wnt4 stimulates the activation of PKC in the cochlea, and that the inhibition of PKC rescues the ectopic Wnt4 activity phenotype. Finally, we demonstrate that modification at a PKC target site on Atoh1 diminishes its ability to induce hair cell formation. Ultimately, we identify a new Wnt/Calcium/PKC non-canonical signaling pathway that is involved in proper hair cell and organ of Corti formation in the developing mammalian cochlea.

## 1. Introduction

Partitioning of epithelial cells into separate functional units consisting of distinct cell types is a fundamental event during the development of any organism or organ system. One example of such partitioning is the coiled mammalian cochlear duct. Initially, the floor of the duct consists of a homogeneous population of epithelial cells [1]. As development progresses, these epithelial cells become segregated into a medial domain of cells that are fated to develop as non-sensory supporting cells, and a lateral domain of cells that become specified to become the auditory sensory epithelium, also called the organ of Corti (OC) [2]. Within the OC, a single row of inner hair cells (IHCs) and three rows of outer hair cells (OHCs) extend along the cochlea, and every sensory hair cell (HC) is separated from the next by an intervening non-sensory supporting cell, resulting in an invariant and alternating mosaic [3,4,5]. The importance of the formation of this structure is illustrated by significant auditory deficits in animals with patterning defects [6,7].

In the developing inner ear, Wnt signaling has been demonstrated to influence proliferation, cell fate, planar cell polarity (PCP), and convergent extension movements of the cochlear duct [8,9,10,11,12,13,14,15,16,17,18,19,20,21,22,23,24,25,26,27,28,29,30,31,32]. Further, many components of the Wnt signaling pathways are expressed in the developing auditory system [10,17,19,22,24,25,33,34,35,36,37,38,39,40,41,42,43,44,45,46,47], suggesting a pleiotropic role for Wnt-Frizzled interactions in various aspects of cochlear development.

Wnt4 is expressed and functional in various systems during organogenesis, including the kidney, Müllerian duct, mammary gland, adrenal gland, and gonads, indicative of multiple roles for this secreted signaling ligand [48,49,50,51,52,53]. The mechanism of Wnt PKC signaling has been well characterized in cell lines; however, less is known about its function in development [54,55,56]. Examples of the *in vivo* role of Wnt PKC signaling include axis formation [57], gastrulation [58], heart field specification [59], synapse formation [60], neuroectoderm patterning [61], and retinal angiogenesis [62]. In particular, Wnt4, one of the many Wnts expressed in the inner ear, has been suggested as a potential candidate in refining and maintaining the boundaries between sensory and non-sensory domains in the chicken inner ear, although functional data for such a role is not yet available [19,37].

In this study, we investigated the role of Wnt4/Calcium/PKC signaling in the developing cochlear duct and provided the first evidence for the involvement of this pathway in regulating the number of HCs within the mammalian cochlea. Collectively, we show that the loss of Wnt4, calcium chelation, and PKC inhibition, all individually, lead to the production of supernumerary ectopic inner hair cells in the cochlea, suggesting the importance of this signaling pathway in the proper regulation of hair cell number in early mammalian cochlear development.

## 2. Materials and Methods

### 2.1. Animals and In Situ Hybridization

Pregnant ICR mice were obtained from Charles River and sacrificed according to UCSD, NIH, and Sunnybrook Research Institute guidelines; embryos were removed, and cochleae were isolated. In situ hybridization was performed on cochleae from mice as described previously [63]. The animal study was reviewed and approved by the Animal Care and Use Committee at Sunnybrook Research Institute (AUP 23-514, approval date: 15 October 2024) and the National Institute on Deafness and Other Communication Disorders, National Institutes of Health.

### 2.2. Analysis of Wnt4 Mutants

*Wnt4^−/−^* mutants were generated by crossing *Wnt4^+/−^* animals [51]. The *Wnt4* null mutation was generated through deletion of the entire *Wnt4* coding sequence, as previously reported [51]. Cochleae from E18.5 and postnatal day (P) 0 mice were isolated, fixed, and HCs were labeled. Animals were bred and maintained in accordance with animal health and care guidelines at the University of Oulu, Finland.

### 2.3. Organotypic Explant Cochlear Cultures

Embryonic mouse stages were determined based on Kaufman 1992 [64]. Cochlear explant cultures from embryonic day (E) 13 or E15 embryos were established, and immunocytochemistry was performed as described previously [65,66]. Antibodies used were mouse αAtoh1 (DSHB), rabbit αMyosin7a (Proteus Bioscience, Ramona, CA, USA), rabbit αMyosin6 (Proteus Bioscience, Ramona, CA, USA), goat αSox2 (Santa Cruz), mouse αBrdU (BD Biosciences, San Jose, CA, USA), goat αWnt 4 (R&D Systems), and goat αsFRP2 (R&D Systems, Minneapolis, MN, USA). BrdU (BD Biosciences, San Jose, CA, USA) was used at a concentration of 3.5 ng/mL and applied contemporaneously with the establishment of the cultures. Culturing procedures and recipes are described in detail with a video tutorial [66].

### 2.4. Modulation of Wnt Signaling

Cochlear cultures were either exposed to media conditioned with Wnt4 protein (NIH3T3 stably transfected with pLNCX-Wnt4 [67]) or to media conditioned by the untransfected NIH3T3 parent cell line. Conditioned media were generated as previously described [10]. Presence of Wnt4 in conditioned media was verified by western blot. sFRP2 (50 µg/mL) was dissolved in the culture media along with 1 µg/mL heparin; control media contained 1 µg/mL heparin [68]. Calcium was chelated using BAPTA-AM (Calbiochem, San Diego, CA, USA). PKC was inhibited using either BIM I or Gö6983. PMA and BIM V were used as positive and negative control compounds, respectively, for PKC inhibition. All compounds were purchased from Calbiochem via Millipore.

### 2.5. PKC Activation Assay

E14 cochleae were dissected and incubated in DMEM for 2 h. Cochleae were treated with Wnt4-CM, control media, or DMEM with 200 nM PMA (Calbiochem) for 30 min. Cochleae were then washed with ice cold PBS and lysed with buffer (10 mM Tris-HCl, pH 7.5, 75 mM NaCl, 0.5% NP-40, 1.25 mM EDTA, 15 mM NaF, 5 mM of Na_2_P_2_O_7,_ 1 mM Na_3_VO_4_, 20 μg/mL aprotinin, 20 μg/mL leupeptin and 1 mM AESBF) for 30 min on ice. Samples were centrifuged at 14,000 rpm for 20 min, and 20 μg of protein per lane was loaded onto a 10% gel for immunoblotting. Gel was transferred onto a nitrocellulose membrane, and the blot was blocked with 5% non-fat milk and then incubated with anti-phospho-PKC (pan) (Cell Signaling) at a dilution of 1:1000 in 5% Bovine serum albumin (BSA) overnight. Anti-PKC (pan) (Calbiochem) and anti-hsp-70 (Santa Cruz Biotechnologies, Dallas, Texas, USA) were used as loading controls, 1:100 and 1:5000, respectively. Secondary antibodies were HRP-conjugated Anti-rabbit IgG (Amersham Biosciences, Piscataway, NJ, USA), used at 1:2000 (anti-phospho-PKC (pan)) and 1:1000 (anti-PKC (pan)). HRP conjugated Anti-mouse IgG (Amersham Biosciences, Piscataway, NJ, USA) was used at 1:10,000. Chemiluminescence was performed with Pierce SuperSignal West Dura. Four independent replicates were performed.

### 2.6. Real Time PCR

Changes in mRNA expression of *Atoh1*, *Hes1*, and *Hes5* were determined by real-time quantitative PCR. We used the multiplex PCR method, allowing the use of more than one primer pair in the same reaction tube. One primer pair amplifies the target (*Atoh1*, *Hes1*, and *Hes5*) gene, and the other primer pair amplifies an internal control—here we used *GAPDH*. Primer pairs and TaqMan probes for *Atoh1*, *Hes1*, and *Hes5* were designed using Primer Express (ABI). GAPDH primers and Taqman probes were purchased from ABI (TaqMan Rodent GAPDH Control Reagents). Three independent experiments were performed on different days, each containing three replicates. E13 cochleae were dissected, cultured, and treated the next day with 2 µM BIM I or DMSO control media for 48 h. RNA was extracted from the cochlear explants using a Total RNA kit (Stratagene, La Jolla, CA, USA). The PCR reaction included cDNA (50 ng), target primers, target TaqMan probe, GAPDH primers, GAPDH TaqMan probe, and TaqMan Universal PCR Master Mix (ABI) in a total volume of 50 μL. Fluorescent emission of the TaqMan probe reporter dye was proportional to amplification of the target gene. We analyzed the data using the comparative Ct method.

### 2.7. Site-Directed Mutagenesis

Putative phosphorylation sites were determined by interrogating the mouse Atoh1 protein reference sequence utilizing NetPhos, Scansite, ExPASy, and kinasePhos internet-based analysis tools for predicting phosphorylation sites. The mutation in the PKC phosphorylation sites in *Atoh1* was created using the QuikChange site-directed mutagenesis kit (Stratagene) following the manufacturer’s instructions. All mutations were confirmed by sequencing.

### 2.8. Transfection

Cochlear explants were transfected using square-wave electroporation as described previously [69]. A plasmid vector using the *pCLIG* backbone to direct expression of *Atoh1* and enhanced green fluorescent protein (*EGFP*) as two independent transcripts (referred to as *pCLIG.Atoh1.EGFP*) was provided by R. Kageyama (Kyoto University, Kyoto, Japan) [70]. In brief, cochlear explants were dissected at E13 and electroporated with either wildtype control (*pCLIG.Atoh1.EGFP*), empty vector negative control (*pCLIG.EGFP*), or either of the two Atoh1 mutants (*pCLIG.Atoh1T197D.EGFP* and *pCLIG.Atoh1S146D.EGFP*). Explants were maintained for 6 days *in vitro* and fixed with 4% paraformaldehyde. HEK293 cells were transfected using FuGENE according to the manufacturer’s instructions.

### 2.9. Statistical Analysis

All experiments were performed on a minimum of three independent occasions (at least three biological replicates) and are specified in the figure legends. All experiments comparing two independent variables were statistically analyzed using two-tailed T-tests. Each experiment was examined using Fisher’s test of equal variance (confidence interval 95%). Where experiments were found to have equal variance, a two-tailed Student’s T-test was performed, testing against a 95% confidence interval. Where two variables had unequal variance, Welch’s T-test for samples with unequal variance was applied using a confidence interval of 95%. Experiments that compared results of more than two groups were examined using ANOVA. Where a significant difference in variance was detected, individual T-tests were performed.

## 3. Results

### 3.1. Wnt4 Is Expressed in the Embryonic Cochlea

The spatiotemporal expression pattern of *Wnt4* was determined at E13, 14.5, and 16.5, representing embryonic stages related to the specification of the prosensory region and the subsequent differentiation of sensory cell types, respectively. An H&E image of the E17 cochlea shows different compartments of the embryonic mouse cochlea (Figure 1A). On E13, prior to the differentiation of hair cells and supporting cells, *Wnt4* is expressed in the medial boundary of the roof of the cochlear duct in an area adjacent to the non-sensory region of the duct (Figure 1B). By E14.5, the floor of the cochlear duct is divided into an actively proliferating medial non-sensory region and a lateral post-mitotic prosensory region, and IHCs have begun to differentiate at the medial edge of the prosensory domain (Figure 1C). *Wnt4* expression is restricted to the medial edge of the roof of the cochlear duct (Figure 1C). By E16.5, individual cells that will develop as a single IHC and three OHCs can be identified within the prosensory region (Figure 1D). It is also evident by this stage that *Wnt4* is expressed in cells that will develop as Reissner’s membrane (Figure 1D). Therefore, the timing of *Wnt4* expression within the cochlea correlates with specification of the sensory and non-sensory regions and with HC fate determination, suggesting that Wnt4 may play a role in the early development of the mammalian cochlea.

### 3.2. Wnt4 Mutants Contain Extra IHCs in the Cochlea

To determine whether Wnt4 plays a role in cochlear development, we used the HC-specific marker Myosin6 to label HCs in cochleae from *Wnt4* homozygous mutants (*Wnt4^−/−^*), Wnt4 heterozygous mice (*Wnt^+/−^*), and wild-type (WT) littermates at P0. In WT cochleae, one row of IHCs and three rows of OHCs were present along the length of the cochlea (Figure 2A). In contrast, cochleae from *Wnt4^−/−^* littermates contained extra IHCs, termed ectopic HCs (arrows in Figure 2B). Higher magnification images show hair cell structure and morphology from both WT and *Wnt4^−/−^* mutant cochleae, respectively (Figure 2A’,B’). Examination of plastic sections from WT and *Wnt4^−/−^* cochleae confirmed the presence of ectopic IHCs in *Wnt4^−/−^* animals (Figure 2C,D). We quantified the occurrence of ectopic IHCs and found a significant increase in the number of ectopic IHCs in *Wnt4^−/−^* cochleae compared to controls (Figure 2E). We observed no isolated ectopic OHCs, nor were there any regions displaying an ectopic row of OHCs. We then counted the OHCs in 100 µm regions along the mid-basal turn of WT, *Wnt4^+/−,^* and *Wnt4^−/−^* cochleae and found no significant change in OHC number between groups (Figure 2F). Furthermore, there was no statistical difference between the total length of WT and mutant cochleae, indicating that the increase in IHCs was not due to a shortened cochlear duct resulting from convergence and extension defects (Figure 2G). Thus, the loss of Wnt4 led to the presence of ectopic IHCs, but not OHCs, in the *in vivo* neonatal cochlea.

### 3.3. Wnt4 Inhibits Hair Cell Formation In Vitro

The increase in IHC number in *Wnt4^−/−^* cochleae suggests a possible inhibitory role for Wnt4 in HC formation; therefore, we investigated the effects of Wnt4-mediated signaling *in vitro* using embryonic cochlear explants. First, we activated the Wnt4-signaling pathway by maintaining cochlear explants in media conditioned by either Wnt4-expressing cells (Wnt4-CM) or untransfected parent cells (control-CM). Western blot confirmed Wnt4 protein expression in Wnt4-CM (Figure 3A). Based on the timing of cell fate decisions in the developing cochlea and Wnt4 spatial expression, explants were established on E13 and exposed to Wnt4-CM or control-CM media for 6 days. Explant cultures maintained in control-CM developed normally, including the formation of the normal cellular pattern of one row of inner and three or four rows of OHCs (Figure 3B,B’). In contrast, E13 explant cultures maintained in Wnt4-CM showed a decrease in the number of Myosin6-positive IHCs and OHCs (Figure 3C,C’). The developmental window for Wnt4-mediated changes in hair cell number is temporally restricted, as the addition of Wnt4-CM to explants established on E15 and incubated for 6 days did not change HC number (Figure 3D,D’). We quantified the inhibitory effect of Wnt4 on HC formation by counting IHCs and OHCs in 100 µm regions at different positions along the basal-to-apical axis of each cochlear explant. There was a significant decrease in the number of IHCs in the Wnt4-CM treatment group at all points measured (Figure 3E). However, the effect was most pronounced in the apical, more immature region of the cochlea (75% from base), suggesting that Wnt4 plays a role in early HC development. Moreover, concentrating Wnt4-CM 10-fold resulted in a more severe effect—a smaller number of cells developed as HCs (Figure 3E). OHCs were similarly affected (Figure 3F), suggesting that the inhibitory effects of Wnt4 are not limited to IHCs and are more severe in embryonic cochlear explants *in vitro*. Hair cell development occurs in a medial-to-lateral gradient where IHCs develop first, followed by row1, row2, and then row3 OHCs [1]. If IHCs do not develop, this leads to the inhibition of OHC development. Therefore, the loss of OHCs observed in Wnt4-CM could be caused indirectly through the drastic loss of IHCs, which is observed in a concentration-dependent manner *in vitro*.

### 3.4. Specificity of Wnt4-Mediated Signaling

Secreted Frizzled-Related Proteins (sFRPs) are secreted Wnt antagonists [71]. Of the five members of the sFRP family, sFRP2 has been shown to bind Wnt4, along with other Wnts, and to inhibit the Wnt4 signaling pathway [72,73,74]. To determine whether sFRP2 could play a role in limiting the range of Wnt4 activity in the cochlea, we localized sFRP2 expression by immunocytochemistry. In contrast to Wnt4, which is expressed in the medial region of the developing cochlea at E13 (Figure 1B and Figure 4A), sFRP2 expression is restricted to the lateral region of the cochlear duct (Figure 4B). We hypothesized that if sFRP2 has an inhibitory effect on Wnt4 signaling in the cochlea, then the application of exogenous sFRP2 to developing cochlear explant cultures would lead to an increase in the number of ectopic HCs. In sFRP2-treated cochlear cultures, there was an increase in the number of IHCs, leading to the formation of almost two complete rows of IHCs compared to control explants containing the normal one row of IHCs (Figure 4C,D). These results are consistent with the hypothesis that the presence of Wnt4, and likely other redundant Wnts, in the developing cochlear duct plays a role in inhibiting the formation of HCs.

### 3.5. Calcium Chelation Results in Ectopic IHCs

We next addressed the question of which intracellular signaling pathway might mediate the effects of Wnt4 in the cochlear duct. While Wnt4 can activate canonical β-catenin-dependent pathways in some specific contexts, our previous work has shown that triggering the accumulation of β-catenin in the E13.5 cochlea results in an increase in the number of HCs [12,17,46]. Since our data here shows that the activation of Wnt4 signaling at E13.5 results in a reduction in the number of HCs (Figure 3), Wnt4 seems unlikely to mediate canonical Wnt signaling in the cochlea at this developmental time point. Moreover, in contrast to Wnt5a, a mediator of Wnt PCP which allows for the coordinated alignment of HCs in the inner ear [14,25], loss of Wnt4 does not result in convergent extension defects in the cochlea (Figure 2G), indicating that Wnt4 does not mediate PCP.

Intriguingly, Wnt4, Wnt5a, and Wnt11 have been shown to activate the Wnt/Calcium/PKC signaling cascade [75]. Briefly, the interaction of a Frizzled trimeric G-protein-coupled receptor with a Wnt ligand triggers calcium release from the endoplasmic reticulum. Calcium accumulation then activates various calcium-sensitive effectors, such as PKC, CamIIK, and NFAT [75]. Based on these results, we sought to determine whether the manipulation of intracellular calcium levels may also influence HC development. If Wnt4 acts through the calcium pathway, then decreasing intracellular calcium levels should result in the formation of supernumerary IHCs, like the ectopic HCs observed following the deletion of *Wnt4 in vivo*. We used a calcium chelating agent, BAPTA-AM, to reduce intracellular calcium concentrations in the developing cochlea. E13.5 cochlear explants were exposed to 20 µM BAPTA-AM (cell permeable) for six days before being fixed and stained with the HC marker Myosin6. HC counts were performed within 100 μm regions at 25%, 50%, and 75% along the length of the duct. As predicted, there was a significant increase in the number of IHCs, but not OHCs, in all regions of BAPTA-AM-treated cochlear explants (Figure 5A–C). This suggests that calcium may be a downstream component of the Wnt4 signaling pathway in the embryonic cochlea.

### 3.6. Inhibition of PKC Induces Ectopic IHCs

As many protein kinase C (PKC) isozymes are activated by calcium release, and the inhibition of both calcium release and Wnt4-mediated signaling results in an increase in IHCs, we hypothesized that the inhibition of PKC should also result in the abrogation of Wnt4 signaling, resulting in the induction of ectopic IHCs. To test this hypothesis, explant cultures were established and treated with the PKC antagonist bisindolymaleimide I (BIM I) beginning on E13. In explant cultures established on E13 and treated with BIM I, there was a robust increase in the number of IHCs (Figure 6A,B). As was the case for Wnt4 treatment, the phenotype of BIM I application was contingent on the developmental stage of the explanted cochleae. BIM I was only able to induce ectopic HCs prior to E15 (Figure 6C). To confirm that these supernumerary HCs were specific to the inhibition of PKC activity and not generalized kinase inhibition, another selective inhibitor of PKC (Gö6983) [76] was applied to E13 cochlear explants. These explants were then compared to explants incubated in a negative control compound that does not inhibit PKC activity effectively (BIM V). Explants in which PKC was selectively inhibited developed ectopic hair cells, whereas explants exposed to BIM V did not (Figure 6D).

Since the inhibition of PKC led to an increase in IHCs, we tested whether there was an increase in mRNA expression of the hair cell master-regulator gene, *Atoh1,* by quantitative real-time PCR. The expression of two supporting cell markers, *Hes1*, predominantly expressed in supporting cells of the IHC region (inner phalangeal/inner border cells [77]) and *Hes5*, predominantly expressed in supporting cells of the OHC region (Deiters’ cells) [78], were also examined. Inhibition of PKC resulted in a significant increase in both *Atoh1* and *Hes1* mRNA expression after 48 h, while no difference was observed in *Hes5* (Figure 6E). The increase in *Hes1* expression likely corresponded to the development of supernumerary supporting cells that form concurrently with new HCs because of Atoh1-initiated Notch1-mediated lateral inhibition [79,80,81]. *Hes5* expression was unchanged, indicating that there was no increase in the number of supporting cells that accompany or surround OHCs. These results were consistent with an effect on IHCs, but not OHCs, and indicate an overall effect specifically on the size of the sensory HC region within the cochlear duct. We then tested whether the ectopic IHCs arose due to the proliferation of IHC precursors. Cultures established on E13 and cultured in media supplemented with BIM I or DMSO and BrdU displayed no significant change in the number of BrdU-labeled HCs in BIM I treated cultures compared to DMSO-treated control cultures (Figure 6F). The absence of cell division in the HC population indicated that the effects of PKC were mediated by direct changes in cell fate within the non-sensory region of the cochlear duct rather than through proliferation.

### 3.7. PKC Is Phosphorylated in Response to the Addition of Wnt4 in the Embryonic Cochlea

We hypothesized that if the Wnt4 signal is transduced through PKC, the effects of Wnt4-CM should be blocked by the inhibition of PKC. To test this hypothesis, cochlear explants were established at E13 and then concomitantly exposed to Wnt4-CM and the PKC inhibitor BIM I. As reported above, explants cultured in Wnt4-CM alone exhibited a decrease in the number of HCs (Figure 7A); however, this phenotype was rescued by application of BIM I (Figure 7B). As would be predicted for a downstream component of the same pathway, BIM I still induced ectopic IHCs (Figure 7B). Thus, the activation of PKC is required for Wnt4-mediated inhibition of HC formation. We next determined whether the phosphorylation of PKC is a target of Wnt4. E14 cochleae were incubated for 30 min with Wnt4-CM, control media, or media treated with phorbol 12-myristate 13-acetate (PMA)—a PKC activator used as a positive control. PKC phosphorylation levels were analyzed by immunoblotting using an anti-phospho-PKC antibody. Wnt4-CM induced a rapid increase in the phosphorylation level, and thus, the activation of PKC in E14 cochleae relative to control media (Figure 7C). In fact, PKC activation in E14 cochleae treated with Wnt4-CM was comparable to that observed in response to the PKC agonist, PMA (Figure 7C), suggesting a specific response. These results are consistent with a direct role for Wnt4 in activation of PKC in the developing cochlea.

### 3.8. Modification of a Phospho-Serine Site on Atoh1 Inhibits Hair Cell Formation

Previous studies have demonstrated that PKC phosphorylation can inhibit the activity of bHLH transcription factors, including *MyoD* and *Myf5* [82,83,84,85,86]. To determine whether the phosphorylation of *Atoh1*, a bHLH transcription factor, could play a role in the regulation of Atoh1, we identified putative PKC phosphorylation sites in the Atoh1 amino acid sequence and used site-directed mutagenesis to create single point mutations (Threonine or Serine to Aspartic Acid) mimicking phosphorylation in those sites. One of the most intriguing putative sites was a threonine located at amino acid site 197, a location near the border between the basic and HLH domains. As a control, serine 146, a putative site located outside the bHLH domain, was also mutated. Since over expression of Atoh1 in the non-sensory region of the cochlea leads to the formation of ectopic hair cells [70,87,88], the effects of each point mutation, as well as empty vector (*pCLIG*) and wildtype controls (*pCLIG-Atoh1*) on Atoh1 activity were assayed by electroporating the mutated *Atoh1* constructs into cochlear explants and assaying for hair cell induction. The empty vector (*pCLIG*) transfected in explant cultures did not induce any HC formation, whereas wild-type Atoh1 (*pCLIG-Atoh1*) led to HC induction (Table 1). Mutation of the phosphorylation site outside the bHLH domain to an aspartic acid residue had no effect on *pCLIG.Atoh1(S146D).EGFP* retained the ability to induce hair cells in non-sensory epithelia (Figure 8A and Table 1).The *pCLIG.Atoh1(T197D).EGFP* construct, in which a Threonine phosphorylation site within the bHLH domain was replaced with an aspartic acid residue to mimic phosphorylation, was not able to induce hair cell formation (Figure 8B and Table 1). To demonstrate the failure of *pCLIG.Atoh1(T197D).EGFP* to induce ectopic HC formation was due to the point mutation, and not a failure of the plasmid to produce protein, we transfected it into HEK293 cells, a cell line that does not express endogenous Atoh1, and assayed for Atoh1 protein by immunocytochemistry, which was found to be expressed after transfection of the plasmid (Figure 8C–F). Thus, HC formation is impaired when phosphorylation of Atoh1 occurs in the bHLH domain, and mutating a phosphorylation site outside the bHLH domain led to HC induction.

Taken together, this study identifies a novel Wnt4/Ca^2+^/PKC non-canonical signaling pathway that, when activated, results in the inhibition of HC formation in the embryonic mouse cochlea. Atoh1 was found to be one of the downstream targets of this pathway, where the proposed pathway suggests that PKC phosphorylates Atoh1 at a putative binding site, which then leads to the inhibition of ectopic HC formation. To our knowledge, this is the first study examining the non-canonical Wnt4/Ca^2+^/PKC signaling pathway in the embryonic mouse cochlea and provides an understanding of the components that are necessary for proper mammalian HC formation in early cochlear development.

## 4. Discussion

### 4.1. Wnt/PKC Modulates Sensory/Nonsensory Cell Fate Decisions in the Greater Epithelial Ridge

The floor of the mammalian cochlear duct is divided into two zones, the medial and lateral compartments. The medial compartment forms the greater epithelial ridge (GER) and will give rise to Kölliker’s organ (which will eventually form the inner sulcus in the mature cochlea), IHCs, inner phalangeal cells, and inner border cells [89,90]. The lateral compartment forms the LER and will give rise to OHCs, surrounding supporting cells, and the lateral nonsensory region, which will later form the outer sulcus [4]. Exquisitely coordinated waves of cell signaling early in development define the compartments of the developing cochlear duct [4,29,91]. While Fgf10 and Fgf20 [92,93] act in combination with BMP4 [94] to specify the lateral compartment, less is known about the specification of the medial compartment. What is clear is that as development progresses, the medial and lateral compartments express unique and specialized molecular profiles [90,95,96,97] that will respond differently to signaling events depending on the underlying combinations of receptors and intracellular effectors specific to each compartment. Here, we show that Wnt4 signaling, through the Wnt/Calcium pathway, acts on prosensory cells of the medial compartment to limit the IHC domain to a single row, thus keeping the cochlear architecture intact. Further, our data suggest that the activation of PKC likely leads to a phosphorylation-dependent modulation of HC-inducing factors, leading to Wnt4-dependent formation of non-sensory cells.

Expansion of the IHC domain in response to the inhibition of PKC was not due to proliferation of HC precursors (Figure 6F), indicating that Wnt4/PKC signaling modulates direct cell fate decisions without mitotic events. During prosensory development, the Sox2/Jagged1 demarcated prosensory region runs along the length of the basal-to-apical cochlear axis centered around the medio-lateral boundary. Interestingly, while the lateral prosensory region, marked by Sox2 and Jagged1 (Jag1), precisely corresponds with the lateral portion of the organ of Corti (OC), the medial Sox2/Jag1 domain extends into Kölliker’s Organ, several cell rows beyond the region that will differentiate as IHCs and associated supporting cells. Upregulation of Notch effectors *Lfng* and *Mfng* in the medial Jag1/Sox2 domain is necessary to limit the number of IHCs [79], suggesting that low Notch signaling may allow Atoh1 to accumulate in cells fated to become IHCs on a stochastic basis. Further, Jag1 regulates medial and lateral compartmentalization by repressing Notch inhibition in lateral supporting cells and inhibiting an OHC fate in the medial compartment [91]. As with PKC inhibition, the disruption of Notch signaling expands the IHC domain, and subsequent Notch-mediated lateral inhibition then generates GER-specific supporting cells [79]. The non-sensory cells of the medial Sox2/Jag1 domain are sensitive to the application of DAPT up to neonatal stages, with inhibition of Notch leading to increased IHC formation [77,98], demonstrating a latent capacity to differentiate as HCs during embryonic and early postnatal stages. Indeed, several studies investigating the regenerative capacity of the mammalian cochlea have shown that supporting cells of the mediolateral boundary and medial GER region are the most common source of induced ectopic HCs [99,100,101,102,103,104].

Fate mapping has demonstrated that a subpopulation of cells that go on to develop as supporting cells express Atoh1 at early stages of sensory epithelium specification [105], giving weight to the hypothesis that an imbalance in Atoh1 levels between prosensory cells and subsequent autoregulatory loops [106] drive the assignment of cell fate in the developing prosensory domain to only a subset of the cells that initially express Atoh1. Therefore, it is possible that in the absence of Wnt4-mediated repression, Atoh1 is not down-regulated in some GER-localized prosensory cells, leading to an overproduction of IHCs in those regions.

Wnt4 is expressed in the mesenchyme during embryonic kidney development and is required to properly develop the kidney epithelium [67]. We were unable to investigate the effect of ectopic IHCs on hearing because *Wnt4* mutants die within 24 h after birth due to renal agenesis, a congenital disability where one or both kidneys fail to develop [51]. At this point, the cochlea is not yet fully developed; thus, no physiological hearing tests can be performed. Whether the production of ectopic HCs seen with the reduction of Wnt4 leads to differences in physiological hearing remains unknown. Additionally, although these ectopic HCs are formed both *in vivo* and *in vitro*, whether the HCs are functional warrants further investigation.

### 4.2. Specificity of Wnt Signaling

Our data suggest not only that Wnt4 generates a negative signaling gradient along the medial-to-lateral axis of the cochlear duct, but that sFRP2, localized on the lateral side of the duct, may produce a counter gradient along the opposite axis. The resulting signals could then be integrated to mediate the specification of the medial sensory/non-sensory boundary. Treatment with sFRP2 induced additional IHCs, consistent with inhibition of Wnt4 signaling. However, we observed a stronger phenotype/effect with application of exogenous sFRP2 than was observed in the *Wnt4* mutants. As we have previously reported, on E12.5, contemporaneous with the initiation of sensory development, Wnts 2b, 4, 5a, 5b, 7a, 7b, 9a, and 11 are expressed in the cochlea [35]. While we have identified Wnt5a and 7b as mediators of stereocilia bundle orientation through PCP [10,14], the roles of Wnt 2b, 5, 5b, 7b, 9b, and 11 in the cochlea are unclear. While one or more of these Wnts might act to orchestrate canonical Wnt signaling that modulates proliferation and differentiation [12,15], multiple Wnts could also act in concert to regulate the medial domain boundary. Since sFRP2 binds several Wnts [71], the stronger phenotype observed in response to sFRP2 treatment could be a result of the inhibition of other Wnts in addition to Wnt4. While application of sFRP2 results in an expansion of the IHC domain, application of sFRP4/5 results in an increase in both IHC and OHCs [11], indicating a degree of selectivity for sFRP activity in the developing OC. Finally, factors that antagonize Wnt4 signaling at the intracellular level could also regulate or refine the initial Wnt4 signal. Neither chelation of calcium nor inhibition of PKC resulted in the production of ectopic OHCs. Given that the GER and LER have characteristic molecular profiles and give rise to different cell types [90,95,97], it is likely that the cells of the LER do not express the necessary intracellular components at sufficient levels to transduce Wnt4-mediated PKC signaling.

*In vitro*, exogenous Wnt4 eliminated the *in vivo* Wnt4 gradient and inhibited the formation of both IHCs and OHCs. It is possible that exogenous Wnt4 can interact with Frizzled receptors in the lateral prosensory region through paracrine signaling and lead to the inhibition of OHC differentiation. Given that Notch-mediated lateral inhibition is propagated by arising IHCs [79,107,108], it is likely that the greater degree of IHC inhibition also contributed to the secondary loss of OHCs, where, if IHCs do not develop, this leads to a halt in the medial to lateral development of OHCs.

### 4.3. Protein Kinase C Regulates Cell Fate in the Cochlear Duct

Major aspects of development are regulated by the phosphorylation states of proteins through the action of protein kinases. One factor playing a central role in cellular signal transduction and growth regulation is PKC, a family of phospholipid-dependent serine-threonine kinases that phosphorylates a variety of target proteins. As discussed, some Wnts, including Wnt4, and Frizzled receptors have been shown to stimulate PKC as part of the non-canonical Wnt/Calcium pathway [109]. Consistent with this, the results presented here identify PKC as a downstream target of Wnt4 signaling and show that the inhibition of this kinase results in the development of ectopic IHCs in the mammalian cochlea. Furthermore, PKC inhibition rescued the phenotype observed in response to exogenous application of Wnt4, suggesting that the PKC pathway is activated in response to Wnt4 treatment. While the results demonstrate activation of PKC in response to Wnt4-treatment, it is important to consider that PKC activation cannot be assumed to be dependent on a single pathway, as its activity can be modulated by several signaling cascades. As a result, PKC kinases can serve as integrators between multiple signaling cascades. Therefore, the PKC-mediated inhibition of IHC formation could reflect the convergence of several different signaling pathways, each of which modulates PKC activity. Consistent with this hypothesis, inhibition of PKC *in vitro* had a more profound effect on the IHC formation than on the elimination of *Wnt4 in vivo*. As discussed, other Wnts could also be acting through PKC, but it is equally possible that other signaling cascades could also be modulating PKC activity. Clearly, the identification and elucidation of other biologically relevant mechanisms regulating this PKC within the developing cochlea are essential for a better understanding of its role in this system.

### 4.4. Post-Translational Modification of Atoh1

*Atoh1*, a bHLH transcription factor, has been shown to be both necessary and sufficient for hair cell formation based on mutant analysis and cell induction assays *in vitro* and *in vivo* [70,87,88,110]. While control of Atoh1 at the transcriptional level by β-catenin [111] and other transcription factors/coactivators [112] is essential for development of various subtypes of neurons, the cerebellum, Merkel cells and auditory hair cells, and development and homeostasis of the lining of the gut, post translational modifications are increasingly being found to play a critical role in the modulation of Atoh1 activity. A key regulator of Atoh1 expression is binding of Atoh1 protein to an E-box domain located in the Atoh1 3′ promoter [106]. The demonstration that Atoh1 positively regulates its own expression suggests that the regulation of Atoh1 protein activity plays a key role in the regulation and maintenance of Atoh1 expression. In the gut and the developing cerebellum, Atoh1 half-life is regulated via phosphorylation-dependent degradation [113]. More recently, phosphorylation of Atoh1 leading to ubiquitin-mediated destruction has proved crucial to the development of the organ of Corti [114,115]. We suggest that the presence of a putative PKC phosphorylation site with the ability to abolish hair cell induction by Atoh1 might contribute yet another layer of regulation. Exploration of the phosphorylation-dependent behavior is currently underway and will undoubtedly prove fascinating. A previous study showed that in one case, mutation of a Serine in the Atoh1 bHLH domain to an Alanine (preventing phosphorylation) led to progressive hearing loss [116]. The modification of this site appears to render Atoh1 inactive as opposed to targeting it for degradation [117]. This case was especially interesting because it demonstrated that the level and duration of Atoh1 activity must be tightly controlled; losing the ability to disrupt Atoh1 activity prevented hair cell maturation.

## 5. Conclusions

In summary, our findings identify for the first time a Wnt4/Ca^2+^/PKC signaling pathway that contributes to the development and patterning of the HCs and OC and provide further proof for the importance of non-canonical Wnt signaling in organ development. While our study demonstrates that Wnt4, through the PKC pathway, contributes to the development of the OC, other components remain to be determined. The exploration of the genes involved in this novel non-canonical Wnt4/Ca^2+^/PKC signaling pathway and their impact on hair cell regeneration strategies should be further investigated.

## Figures and Tables

**Figure 1 cells-14-00888-f001:**
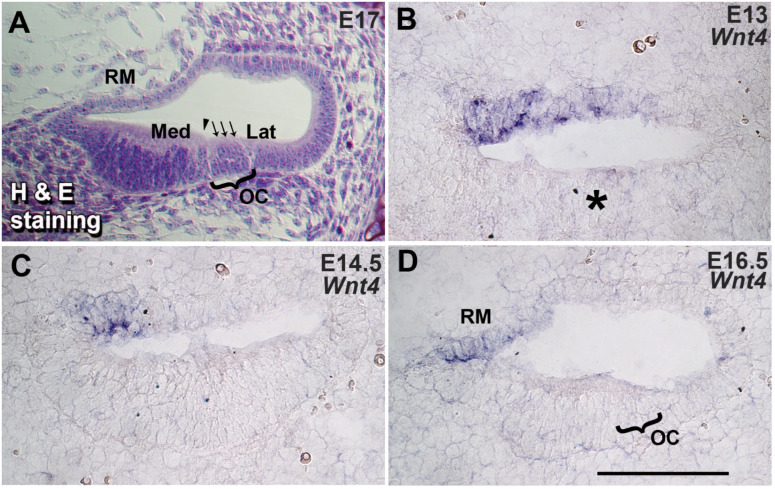
*Wnt4* is expressed in the developing cochlea. (**A**) Semi-thin cross-section through the basal turn of an E17 cochlea stained with hematoxylin and eosin (H&E) to illustrate the composition of a wild-type developing cochlear duct. A single IHC (arrowhead) and three OHCs (three arrows) can be identified in the organ of Corti (OC). In all cross-sections, medial (modiolar, Med) is toward the left, and lateral (strial, Lat) is toward the right. RM: Reissner’s membrane. (**B**) In situ hybridization for *Wnt4* in a cross-section through the basal turn of an E13 cochlea. *Wnt4* is expressed in the medial half of the roof of the cochlear duct. The expression of *Wnt4* is less than 50 µm from the prospective OC (indicated by an asterisk). (**C**) Cross-section through the basal turn of an E14.5 cochlea. *Wnt4* expression persists in the medial region of the roof of the cochlear duct. (**D**) Cross-section through the basal turn of an E16.5 cochlea. *Wnt4* expression is restricted to cells at the medial edge of the developing Reissner’s membrane. The scale bar represents 50 µm.

**Figure 2 cells-14-00888-f002:**
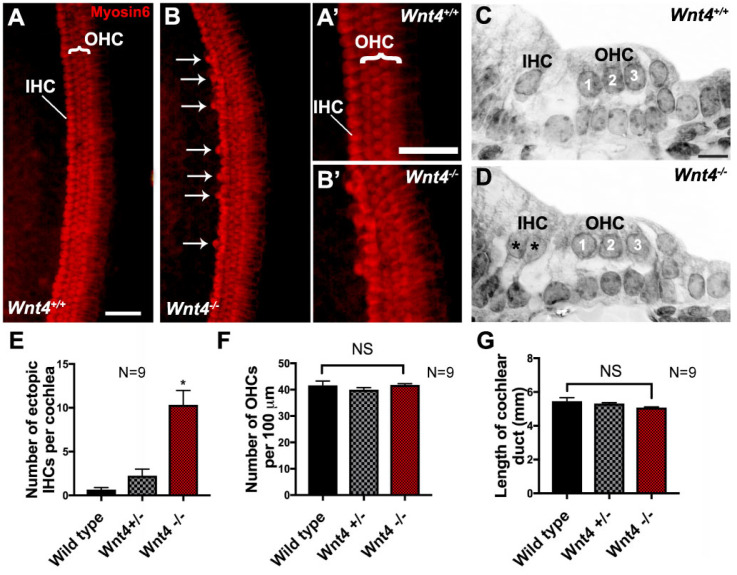
*Wnt4* is necessary for limiting the number of inner hair cells *in vivo*. (**A**) Whole mount surface view of a P0 cochlea (mid-base) from a wild-type mouse with the normal pattern of one row of IHCs and three rows of OHCs labeled with anti-Myosin6. (**B**) Surface view of the mid-base of a P0 *Wnt4*^−/−^ cochlea. In addition to the one row of IHCs and three rows of OHCs, extra IHCs (arrows) are present along the length of the OC. (**A’**) and (**B’**) Zoomed in confocal images of (**A**) and (**B**), respectively. (**C**) Cross-section of the OC from a P0 wild-type mouse. The normal pattern of one IHC and three OHCs (1, 2, 3) is present. (**D**) Cross-section of the OC from a P0 *Wnt4^−/−^* cochlea. Two IHCs (asterisks) and three OHCs (1, 2, 3) are present. (**E**) Average number of ectopic IHCs in P0 cochleae from WT, *Wnt4^+/−^*, and *Wnt4^−/−^* mice. A significant increase in the number of IHCs was present in *Wnt4^−/−^* cochlea as compared to wild-type littermates (Welch’s two-tailed T-test, *n* = 9 for each genotype; *p* = 0.0003). Asterisks indicate statistical significance (*). (**F**) Average number of OHCs in cochleae from P0 WT, *Wnt4^+/−^*, and *Wnt4^−/−^* mice. There was no significant (NS) change in the number of OHCs in cochleae from any of the three genotypes in a 100 µm region along the mid-base of the cochleae (ANOVA; *p* = 0.4, F = 0.87). (**G**) There was no significant difference in the lengths of the wild type and *Wnt4^−/−^* cochleae, indicating that there is no convergence and extension phenotype (ANOVA, *p* = 0.18, F = 2.3). Error bars indicate SEM. Scale bar in (**A**) and (**A’**) represents 20 µm, scale bar in (**C**) represents 10 µm.

**Figure 3 cells-14-00888-f003:**
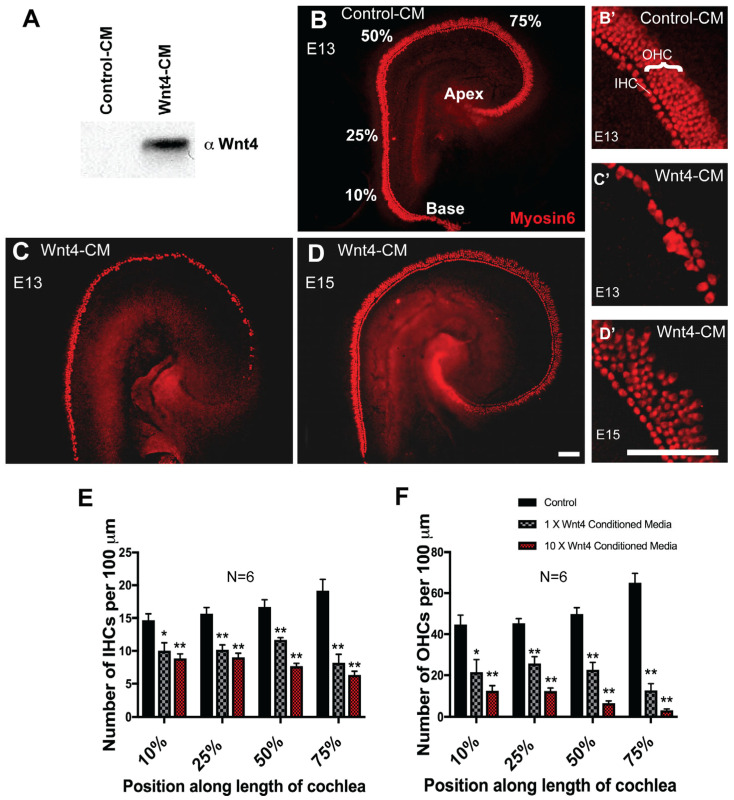
Wnt4 inhibits hair cell formation *in vitro*. (**A**) Western blot showing that the Wnt4-conditioned media (Wnt4-CM) contained Wnt4 protein. (**B**) E13 cochlear explant cultured for 6 days *in vitro* in control-conditioned media (Control-CM). A single row of IHCs and three rows of OHCs (labeled with anti-Myosin6) are present. Percentages indicate the percentage distance from the base of the OC. (**C**) Explant treated with Wnt4-CM beginning at E13. The number of HCs is clearly reduced, especially in the apical region. (**D**) Treatment with Wnt4-CM beginning at E15. (**E**) Wnt4 inhibits IHC formation. (**B’**), (**C’**), and (**D’**) are zoomed-in confocal images of (**B**), (**C**), and (**D**), respectively, from the apical (approximately 75%) region of the explant. The number of IHCs was determined for a 100 µm length of the OC at the positions indicated. A significant reduction in the number of cells that developed as IHCs was observed in cochleae treated with 1× or 10× Wnt4-CM at each position (Student’s test, two-tailed, *n* = 6 for each treatment condition. * *p* < 0.02, ** *p* < 0.002). (**F**) Wnt4 inhibits OHC formation. Wnt4-CM resulted in a significant reduction in the number of cells that developed as OHCs along the length of the cochlea (*n* = 6; * *p* < 0.02, ** *p* < 0.0007). Error bars indicate SEM. Scale bars represent 50 µm.

**Figure 4 cells-14-00888-f004:**
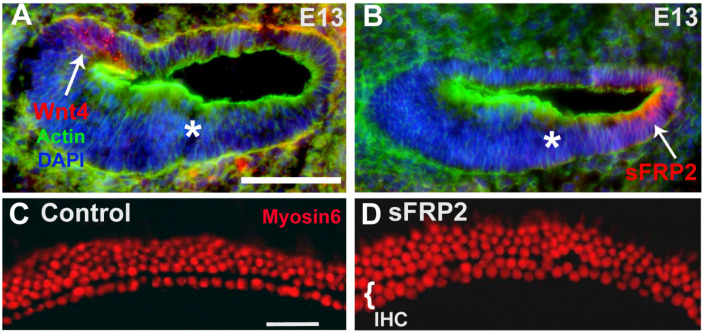
Wnt antagonist sFRP2 is expressed contemporaneously with Wnt4 and has the reverse effect of Wnt4. (**A**) Cross-section through the cochlear duct at E13 illustrating the expression of Wnt4 (red, arrow) protein by immunohistochemistry. The asterisk indicates the prospective organ of Corti. (**B**) Similar cross-section as in A, showing sFRP2 expression by immunohistochemistry. sFRP2 (red, arrow) is expressed on the lateral edge of the cochlear duct—lateral to the area where the OC (*) will develop. (**C**) Surface view of the mid-base of a control cochlear explant culture established on E13 and maintained for 6 days *in vitro*. A single row of IHCs and three to four rows of OHCs are labeled with anti-Myosin6 (red). (**D**) Similar view as in C from an explant established on E13 and treated with 50 µg/mL sFRP2. An almost continuous second row of IHCs is present. Scale bars represent 50 μm.

**Figure 5 cells-14-00888-f005:**
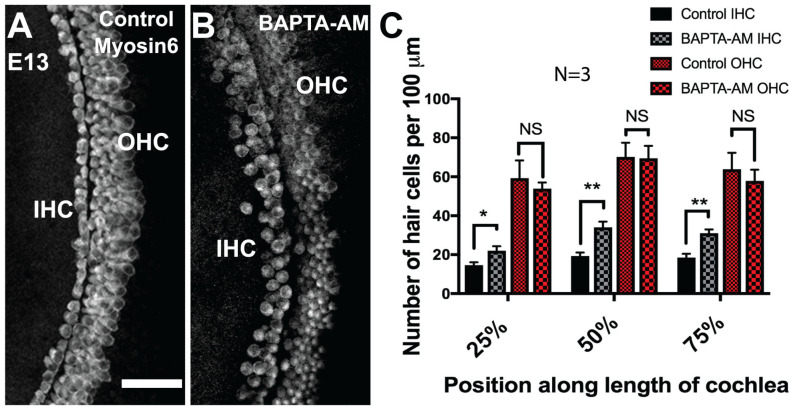
Calcium chelation induces ectopic IHCs in the developing organ of Corti. (**A**) Surface view of the mid-base of an explant culture established at E13 and maintained in control DMSO media for 6 days, the typical three rows of OHCs and one row of IHCs are present. (**B**) Surface view of the mid base of an explant culture established on day E13 and maintained in media containing 20 µM of the calcium chelator BAPTA-AM. There is an increase in the number of cells that develop into IHCs. (**A**) and (**B**) show the mid-base of the cochlea; hair cells are labeled with anti-Myosin6. (**C**) Quantification of the increase in IHC and OHC number in cultures treated with BAPTA-AM. Cell counts were taken at 25%, 50%, and 75% of the length of the explant. (*) *p* < 0.05; (**) *p* < 0.001; 3 independent experiments for each treatment condition. Bars indicate SEM. The scale bar represents 50 μm.

**Figure 6 cells-14-00888-f006:**
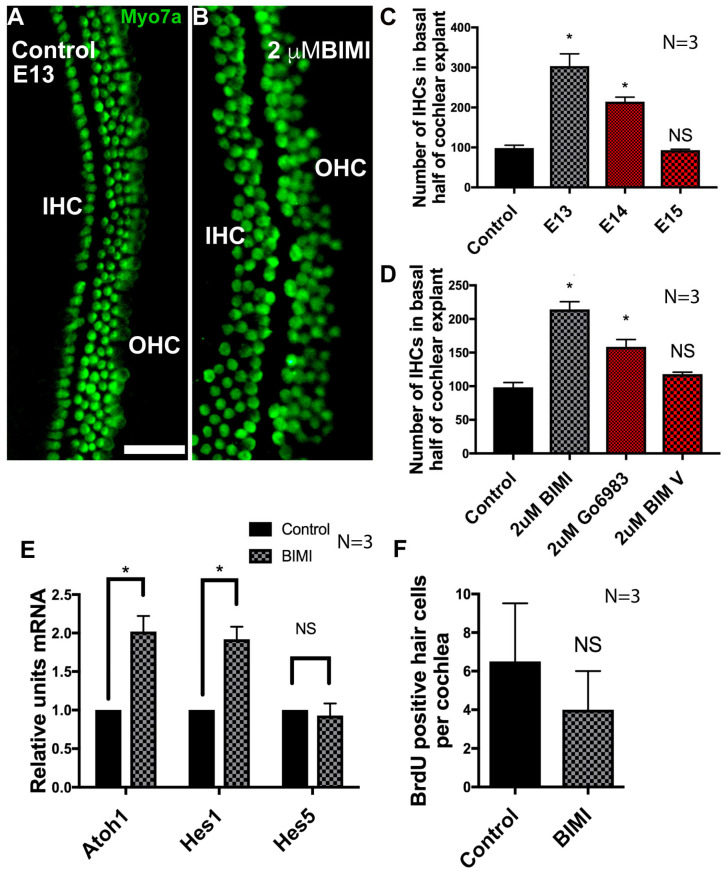
Inhibition of PKC induces the formation of ectopic IHCs in the developing organ of Corti. (**A**) An explant organ culture established at E13 and maintained in control media for 6 days. The typical three rows of OHCs and one row of IHCs are present. (**B**) An explant culture established at E13 treated with the PKC inhibitor BIM I (2 µM). Supernumerary IHCs have differentiated. Images (**A**) and (**B**) show the mid-base of the cochlear cultures, HCs labeled with anti-Myosin7a. (**C**) Quantification of the increase in IHCs in response to the activation of PKC at different stages of development. BIM I was added to explant cultures at E13, E14, and E15. Relative to control E13 explants cultured in DMSO (*n* = 3), there was a significant increase in the number of IHCs when cultures were exposed to BIM I beginning on E13 (*n* = 3) and 14 (*n* = 3); this effect was abolished by E15 (*n* = 3). Error bars represent SEM, * *p* < 0.05. (**D**) Quantification of the increase in IHCs in response to the inhibition of PKC. Two highly selective PKC inhibitors, BIM I (*n* = 3) or Gö6983 (*n* = 3), significantly increased the number of IHCs compared to explants cultured in DMSO (*n* = 3). BIM V, used as a negative control, had no significant effect. * *p* < 0.01. (**E**) Quantification of *Atoh1*, *Hes1*, and *Hes5* expression levels in control and BIM I treated cochlear explant cultures by real-time quantitative PCR. In BIM I treated cultures, *Atoh1* was expressed 2.02-fold more than DMSO treated control cultures after 48 h of treatment, *p* = 0.007. *Hes1* was elevated 1.97-fold, *p* = 0.005. *Hes5* expression levels were unchanged (fold change was 0.97). Control *n* = 3. BIM I treated *n* = 3. Asterisks (*) denote statistical significance. The scale bar represents 50 µm. (**F**) Quantification of BrdU-positive hair cells (*p* = 0.5) in explant cultures treated with BIM I, concurrent with the expansion in the number of inner hair cells. *n* = 3, error bars indicate SEM, NS = no significance.

**Figure 7 cells-14-00888-f007:**
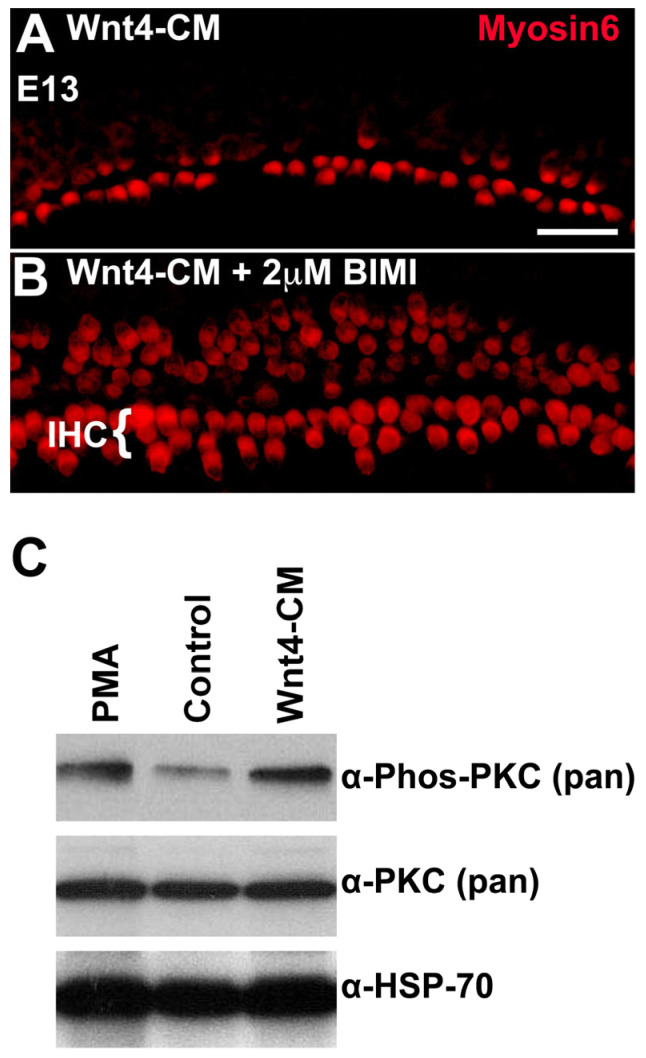
PKC, activated by Wnt4, modulates cell fate. (**A**) An image of a region in the mid-base of an explant established and treated with Wnt4-CM beginning at E13. As illustrated in Figure 2, the number of HCs is reduced in the presence of Wnt4. (**B**) Explant from mid-base treated with Wnt4-CM and the PKC inhibitor BIM I (2 µM). Inhibition of PKC not only blocks the inhibitory effect of Wnt4 but results in the formation of extra IHCs, as was observed in the presence of BIM I alone. (**C**) Wnt4 activates PKC in cochlear tissue. Phosphorylation of PKC was determined in E14 cochleae, untreated, treated with the PKC activator PMA, or treated with Wnt4-CM. The total level of PKC, as assayed using the pan-PKC antibody, was unchanged between the three treatments; however, the levels of phospho-PKC were increased in cochleae treated with PMA or Wnt4-CM. HSP-70 levels were assayed as a loading control. The scale bar represents 50 µm.

**Figure 8 cells-14-00888-f008:**
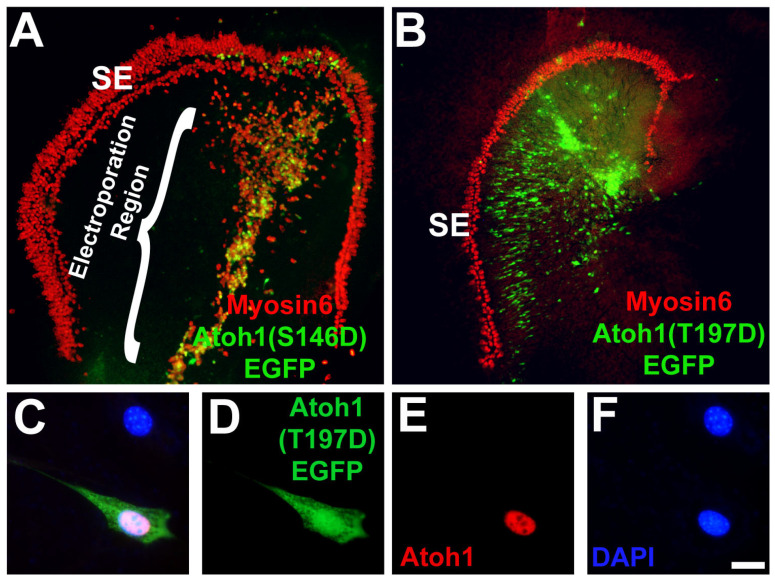
Atoh1 function is ablated by phosphorylation of a PKC target site. (**A**) Cochlear explant culture established on E13 after 6 days *in vitro*. Hair cells are labeled with anti-Myosin6 (red). Non-sensory epithelial cells located medial to the sensory epithelium (SE) have been transfected with the Atoh1(S146D).EGFP mutant. Expression of Atoh1(S146D) induces the formation of ectopic hair cells (yellow cells). (**B**) Cochlear explant transfected with the Atoh1(T197D).EGFP mutant showed no induced hair cells. Note that none of the green cells are positive for expression of Myosin6 (red). (**C**–**F**) Overexpression of Atoh1(T197D).EGFP results in Atoh1 protein formation in HEK293 cells. (**C**) Two HEK293 cells marked by DAPI (blue), one of which has been transfected with Atoh1(T197D).EGFP (green). The transfected cell is positive for Atoh1 protein (red). (**D**–**F**) show individual channels. Scale bars represent 10 µm.

**Table 1 cells-14-00888-t001:** Results of pClig-Atoh1 functional assay.

pCLIG	pCLIG-Atoh1	pCLIG-Atoh1 (S146D)	pCLIG-Atoh1 (T197D)
0% HC	98.5% HC	97.1% HC	0% HC
*n* = 57	*n* = 204	*n* = 340	*n* = 195

HC = Hair cell.

## Data Availability

The original contributions presented in this study are included in the article. Further inquiries can be directed to the corresponding author.

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
