# Peer review of "Wnt/PKC Signaling Inhibits Sensory Hair Cell Formation in the Developing Mammalian Cochlea"

_cells, 2025, doi:10.3390/cells14120888_

Round 1

Reviewer 1 Report

Comments and Suggestions for Authors

The manuscript by Mulvaney et al. provides the first direct functional evidence in mammals that Wnt4, via the non-canonical Wnt/Calcium/PKC pathway, acts to inhibit sensory Inner hair cell formation during cochlear development. Previous studies have suggested roles for Wnt signaling in cochlear patterning, proliferation, and planar cell polarity; however, these have focused mainly on canonical (β-catenin-dependent) pathways. The identification of a Wnt4/PKC pathway that restricts IHC formation adds an important layer to our understanding of cochlear development and patterning.

The experimental design is comprehensive, using genetic (Wnt4 knockout), pharmacological (PKC inhibitors), and molecular (Atoh1 mutagenesis) approaches. The results are robust, with appropriate controls and statistical analysis. The manuscript will attract readers interested in developmental biology and auditory neuroscience.

Some minor limitations and / or elements of discussion may include:

  • The use of WNT4-conditioned media is a practical approach to study WNT4 signaling in cochlear explants. However, the absence of quantitative titration or measurement of WNT4 concentration in the conditioned media introduces uncertainty regarding the effective ligand dose. This lack of quantification may affect reproducibility and limits the ability of other researchers to build upon these findings or perform dose-response analyses. Future experiments should consider using defined concentrations of purified ligand (if commercially available) or, at minimum, quantifying WNT4 levels in conditioned media (e.g., by ELISA or quantitative Western blot).
  • The observation that sFRP2, a broad-spectrum Wnt inhibitor, produces a more pronounced phenotype than WNT4 deletion suggests that other Wnt ligands or parallel signaling pathways may compensate for the loss of WNT4. Additional experiments-such as profiling the expression of other Wnt ligands in the cochlea, or using combinatorial genetic or pharmacological inhibition-would help clarify the extent of compensation and the specificity of the observed effects.
  • The study uses two PKC inhibitors, BIM I and Gö6983, to demonstrate the role of PKC in IHC development. Notably, BIM I is known to inhibit GSK3β at concentrations lower than those used in the study, raising the possibility that canonical Wnt signaling could also be affected. The observed differences in potency to induce IHC between BIM I and Gö6983 may reflect differences in target specificity, including effects on GSK3β or other kinases. It would be interesting to have this element discussed in the paper.
  • The identification of a putative PKC phosphorylation site on Atoh1 is based on in silico prediction tools, and the functional analysis relies on mutagenesis of this predicted site. However, direct biochemical evidence that PKC phosphorylates Atoh1 at this site in vivo is missing. This limits the strength of the mechanistic claim. Future studies should aim to provide direct biochemical validation.
  • Figure 1 is difficult to read and should be enlarged for clarity.

Author Response

  • The use of WNT4-conditioned media is a practical approach to study WNT4 signaling in cochlear explants. However, the absence of quantitative titration or measurement of WNT4 concentration in the conditioned media introduces uncertainty regarding the effective ligand dose. This lack of quantification may affect reproducibility and limits the ability of other researchers to build upon these findings or perform dose-response analyses. Future experiments should consider using defined concentrations of purified ligand (if commercially available) or, at minimum, quantifying WNT4 levels in conditioned media (e.g., by ELISA or quantitative Western blot).

We thank the reviewer for the excellent suggestion that can be implemented in future studies as suggested.

  • The observation that sFRP2, a broad-spectrum Wnt inhibitor, produces a more pronounced phenotype than WNT4 deletion suggests that other Wnt ligands or parallel signaling pathways may compensate for the loss of WNT4. Additional experiments-such as profiling the expression of other Wnt ligands in the cochlea, or using combinatorial genetic or pharmacological inhibition-would help clarify the extent of compensation and the specificity of the observed effects.

Thank you for your comment. We agree that there may be compensation of other Wnts that are expressed in the cochlea that could compensate for the loss of Wnt4. Our laboratory has previously shown that Wnts 2b, 4, 5a, 5b, 7a, 7b, 9a and 11 are expressed in the cochlea and we highlight this possibility in the discussion (line 594).

Geng, R., et al., Comprehensive Expression of Wnt Signaling Pathway Genes during Development and Maturation of the Mouse Cochlea. PLoS One, 2016. 11(2): p. e0148339. (reference 35 in manuscript).

  • The study uses two PKC inhibitors, BIM I and Gö6983, to demonstrate the role of PKC in IHC development. Notably, BIM I is known to inhibit GSK3β at concentrations lower than those used in the study, raising the possibility that canonical Wnt signaling could also be affected. The observed differences in potency to induce IHC between BIM I and Gö6983 may reflect differences in target specificity, including effects on GSK3β or other kinases. It would be interesting to have this element discussed in the paper.

Thank you for your comment. Although the canonical Wnt pathway could be activated at the same time, we and others have shown that at this developmental time period the canonical Wnt pathway involves proliferation that leads to an increase of both types of HCs. In this study we show that there is no proliferation of hair cells (Figure 6), making it unlikely that the canonical Wnt/bcatenin pathway is involved in this process. Furthermore, some of the differences between the two PKC inhibitors could be explained by the differences in potency and specificity.

  • The identification of a putative PKC phosphorylation site on Atoh1 is based on in silico prediction tools, and the functional analysis relies on mutagenesis of this predicted site. However, direct biochemical evidence that PKC phosphorylates Atoh1 at this site in vivo is missing. This limits the strength of the mechanistic claim. Future studies should aim to provide direct biochemical validation.

We thank the reviewer for the suggestion that can be implemented in future studies.

  • Figure 1 is difficult to read and should be enlarged for clarity.

Thank you for this suggestion. We have updated and enlarged Figure 1 for clarity.

Reviewer 2 Report

Comments and Suggestions for Authors

The manuscript reveals a novel mechanism by which Wnt4 inhibits the formation of sensory hair cells in the mammalian cochlea through the non-canonical Wnt/calcium/PKC signaling pathway, demonstrating a certain degree of innovation. Previous studies have predominantly focused on other aspects of Wnt signaling in cochlear development, such as cell proliferation and cell polarity. In contrast, this manuscript zeroes in on the inhibitory mechanism of hair cell formation, offering a fresh perspective on the fine regulation of cochlear development. However, the manuscript still has the following problems:

Major problems:

  1. Figure 6a only shows the changes in hair cells treated with BIMI at E13, but Figure 6c includes statistics for E13, E14, and E15. The manuscript should supplement the hair cell counting immunofluorescence images for E14 and E15.
  2. Figure 6d presents a bar chart summarizing the counts of inner hair cells after various treatments, but it lacks the corresponding immunofluorescence images for hair cell counting.
  3. Figure 6f is missing the immunofluorescence images for BrdU staining.
  4. It is unclear whether BAPTA-AM treatment of embryonic cochlear cultures could be toxic to cochlear cells. It is recommended to include experiments assessing the impact of BAPTA-AM on the survival rate and health status of cochlear cells.
  5. The exploration of the specific molecular mechanisms downstream of the Wnt4/PKC pathway is still insufficient. What changes occur to Atoh1 after its phosphorylation by PKC? Does it interact with other transcription factors?
  6. Does the Wnt4/Calcium/PKC signaling pathway interact with other signaling pathways, such as the canonical Wnt/β-catenin pathway and the Notch signaling pathway, to influence the development of hair cells?

Minor problems:

  1. The language of the manuscript should be simplified, and the logical structure should be optimized.
  2. The labels on the manuscript’s figures and tables could be more detailed. For bar charts showing changes in hair cell numbers, the specific sample sizes for each group should be included.
  3. The authors are strongly encouraged to cite the most recent pertinent literature to ensure the completeness and currency of the review. It is recommended to cite the following relevant references to enhance the scientific rigor of the article.

(1).The Effects of Viral Infections on the Molecular and Signaling Pathways Involved in the Development of the PAOs.

(2).FOXG1 promotes aging inner ear hair cell survival through activation of the autophagy pathway..

  1. In the discussion section, further exploration of hair cell regeneration strategies based on the Wnt4/PKC signaling pathway is encouraged.
Comments on the Quality of English Language

The English language can be improved.

Author Response

  1. Figure 6a only shows the changes in hair cells treated with BIMI at E13, but Figure 6c includes statistics for E13, E14, and E15. The manuscript should supplement the hair cell counting immunofluorescence images for E14 and E15.

We thank the reviewer for their careful assessment. While Figure 6A and B highlight representative immunofluorescence images of E13 hair cells treated with BIMI (a time point where phenotypic changes are most pronounced), the statistical analysis in Figure 6C aggregates data across all three developmental stages (E13–E15) to demonstrate the developmental window of PKC inhibition.

  1. Figure 6d presents a bar chart summarizing the counts of inner hair cells after various treatments, but it lacks the corresponding immunofluorescence images for hair cell counting.

Thank you for your comment. This approach allows us to present the full dataset and statistical comparisons across all treatments, ensuring that the results are robust, specific and statistically significant.

  1. Figure 6f is missing the immunofluorescence images for BrdU staining.

Thank you for your comment. We believe this provides clear and rigorous evidence for the observed effects while keeping the focus on the main point of the figure.

  1. It is unclear whether BAPTA-AM treatment of embryonic cochlear cultures could be toxic to cochlear cells. It is recommended to include experiments assessing the impact of BAPTA-AM on the survival rate and health status of cochlear cells.

Thank you for your comment. Figure 5 shows the hair cell counts of both inner and outer hair cells with BAPTA-AM treatment. We found that BAPTA-AM led to ectopic inner hair cells while no differences in outer hair cell counts were observed compared to controls. If BAPTA-AM was toxic to cochlear cells we would expect to see a reduction in hair cell numbers. Since BAPTA-AM treatment leads to extra IHCs (Figure 5), we conclude that BAPTA-AM itself is not toxic to these cochlear cells.

  1. The exploration of the specific molecular mechanisms downstream of the Wnt4/PKC pathway is still insufficient. What changes occur to Atoh1 after its phosphorylation by PKC? Does it interact with other transcription factors?

Thank you for your comment. The changes that occur to Atoh1 after its phosphorylation by PKC and its potential interactions with other transcription factors are unknown at this time and beyond the scope of the current study. Future studies can examine the downstream regulatory network of this pathway.

  1. Does the Wnt4/Calcium/PKC signaling pathway interact with other signaling pathways, such as the canonical Wnt/β-catenin pathway and the Notch signaling pathway, to influence the development of hair cells?

The current study describes a novel Wnt4/Calcium/PKC non-canonical signalling pathway in the developing cochlea which has not yet been described. Thus, it is currently unknown whether this pathway interacts with other signaling pathways such as the canonical Wnt/b-catenin and/or Notch pathway to influence hair cell development. Future studies are needed to expand our knowledge of this pathway and potential interactions.

Minor problems:

  1. The language of the manuscript should be simplified, and the logical structure should be optimized.

Thank you for your comment. We have revised some figures and text within the manuscript for further simplification and clarity.

  1. The labels on the manuscript’s figures and tables could be more detailed. For bar charts showing changes in hair cell numbers, the specific sample sizes for each group should be included.

Thank you for your suggestion. We have updated Figures 2, 3, 5, and 6 to include the sample size in bar chart panels. The figure legends were also updated to include “N=# for each experimental condition” for clarity.

  1. The authors are strongly encouraged to cite the most recent pertinent literature to ensure the completeness and currency of the review. It is recommended to cite the following relevant references to enhance the scientific rigor of the article.

(1).The Effects of Viral Infections on the Molecular and Signaling Pathways Involved in the Development of the PAOs.

(2).FOXG1 promotes aging inner ear hair cell survival through activation of the autophagy pathway.

Thank you for the additional updated references. We have added “Liu et al. 2024, The effects of viral infections on the molecular and signalling pathways involved in the development of the PAOs” in the introduction, line 45.

We have not added the reference from “He et al. 2021, Foxg1 promoted aging inner ear hair cell survival through activation of the autophagy pathway” as this study looks at the regulatory function of Foxg1 through autophagy during hair cell regeneration in presbycusis and does not examine Wnt, thus it is not relevant to our study.

  1. In the discussion section, further exploration of hair cell regeneration strategies based on the Wnt4/PKC signaling pathway is encouraged.

Thank you for your suggestion. We have added the following “Exploration of the genes involved in this novel non-canonical Wnt4/Ca2+/PKC signaling pathway and their impact on hair cell regeneration strategies should be further investigated” in line 685 to emphasize this point in the conclusion.

Reviewer 3 Report

Comments and Suggestions for Authors

This study investigates the effects of manipulating Wnt/PKC signaling and calcium on cochlear hair cell formation in mice. The authors show that Wnt4 mostly regulates the fate of inner hair cells in a manner that may be dependent on PKC and calcium. The further show that the activity of Atoh1 in specifying hair cell fate may be regulated by PKC-mediated phosphorylation. These observations are potentially interesting and should contribute to understanding the mechanism underlying hair cell formation. However, there are several major issues that need to be clarified before the manuscript can be considered for publication.

  1. In several figures, for example figures 2 and 3, the rows of IHC and OHC are barely visible. To improve the quality and increase the impact of this work, I suggest using confocal and higher magnification images to show clearly the organization of IHCs and OHCs in different conditions.
  2. Figure 8 and its legend are confusing. The legend does not correspond to the images, and no transfection of wild-type Atoh1 is shown. Similarly, it is not clear whether the amino acid at position 197 of the Atoh1 protein is a threonine or a serine (see lines 406, 413, and 419). HEK293 cells do not express Atoh1, while transfected Atoh1 is accumulated both in the cytoplasm and nucleus, as seen by the GFP signals. However, why does Atoh1 immunofluorescence staining detect its presence only in the nucleus?
  3. In Table 1, no result for Atoh1 (T197D) is shown.
  4. There is no clear evidence that modification of Atoh1 phosphorylation inhibits hair cell formation (line 406). The authors only show that transfection of phosphorylation modified Atoh1 (T197D) does not induce ectopic hair cells. They should demonstrate that Atoh1(T197D) acts as dominant mutant to inhibit the formation of IHCs and OHCs.
  5. The authors conclude that ‘PKC phosphorylates Atoh1 at a putative binding site which then leads to inhibition of HC formation’. However, there is no evidence demonstrating that Atoh1 is phosphorylated by PKC. They should at least examine whether inhibition of PKC affects the phosphorylation of Atoh1.
  6. Depletion of Wnt4 affects IHCs but not OHCs. However, exogenous Wnt4 treatment affects both IHCs and OHCs. How the loss of IHCs indirectly affects OHCs (lines 241-243)? In the discussion section, the authors need to further elaborate how Wnt4 interaction with Frizzled receptors inhibits both IHCs and OHCs.
  7. There is direct evidence that Wnt4 inhibits hair cell formation through regulation of calcium concentrations. Can calcium chelation rescue the loss of hair cells caused by exogenous Wnt4?

Author Response

  1. In several figures, for example figures 2 and 3, the rows of IHC and OHC are barely visible. To improve the quality and increase the impact of this work, I suggest using confocal and higher magnification images to show clearly the organization of IHCs and OHCs in different conditions.

Thank you for your suggestion. We have updated Figure 2 and Figure 3 to include zoomed in areas from confocal images for each condition that provide a higher magnification view. We highlight the IHC and OHC regions so their organization can be better viewed in the figure for enhanced clarity.

  1. Figure 8 and its legend are confusing. The legend does not correspond to the images, and no transfection of wild-type Atoh1 is shown. Similarly, it is not clear whether the amino acid at position 197 of the Atoh1 protein is a threonine or a serine (see lines 406, 413, and 419). HEK293 cells do not express Atoh1, while transfected Atoh1 is accumulated both in the cytoplasm and nucleus, as seen by the GFP signals. However, why does Atoh1 immunofluorescence staining detect its presence only in the nucleus?

Thank you for pointing this out. We have updated Figure 8 and its legend.

Transfection of wildtype Atoh1 into cochlear explants has been performed in both the current and previous studies and induces HC formation (Table 1 in manuscript). We have edited the manuscript to further elaborate on the controls used in line 460 “Since over expression of Atoh1 in the non-sensory region of the cochlea leads to the formation of ectopic hair cells [70, 86, 87], the effects of each point mutation, as well as empty vector (pCLIG) and wildtype controls (pCLIG-Atoh1) on Atoh1 activity were assayed by electroporating the mutated Atoh1 constructs into cochlear explants and assaying for hair cell induction.”

We have also edited the text to clarify the results of the wildtype Atoh1 transfection starting at line 464.

“The empty vector (pCLIG) transfected in explant cultures did not induce any HC formation whereas wildtype Atoh1 (pCLIG-Atoh1) led to HC induction (Table 1).”

The amino acid at position 197 of Atoh1 is a Threonine (T197D) and this has been edited in the manuscript (line 462).

The Atoh1 construct used in Figure 8 with HEK293 cells is not directly tagged with GFP but uses an internal ribosomal entry site, therefore, the GFP is expressed throughout the cell, including the cytoplasm as shown in Figure 8. Immunofluorescent staining of Atoh1 labels the nuclei as Atoh1 is a transcription factor.

  1. In Table 1, no result for Atoh1 (T197D) is shown.

Thank you for pointing this out as this was an error on our part. We have added the results for Atoh1 (T197D) in Table 1 of the manuscript (line 510).

  1. There is no clear evidence that modification of Atoh1 phosphorylation inhibits hair cell formation (line 406). The authors only show that transfection of phosphorylation modified Atoh1 (T197D) does not induce ectopic hair cells. They should demonstrate that Atoh1(T197D) acts as dominant mutant to inhibit the formation of IHCs and OHCs.

Thank you for your comment. We show that transfection of phosphorylated modified Atoh1 (T197D) does not induce ectopic HCs whereas transfection of phosphorylated Atoh1 that lies outside the bHLH domain (S146D) led to the formation of ectopic HCs (Figure 8, Table 1). The question of whether Atoh1(T197D) acts as a dominant mutant can be investigated in future studies.

  1. The authors conclude that ‘PKC phosphorylates Atoh1 at a putative binding site which then leads to inhibition of HC formation’. However, there is no evidence demonstrating that Atoh1 is phosphorylated by PKC. They should at least examine whether inhibition of PKC affects the phosphorylation of Atoh1.

Thank you for your comment. We have edited this in the manuscript to “…the proposed pathway suggests that PKC phosphorylates Atoh1 at a putative binding site which then leads to inhibition of ectopic HC formation” (line 482) for better clarity and depiction of the current study. Future studies can examine whether inhibition of PKC affects phosphorylation of Atoh1.

  1. Depletion of Wnt4 affects IHCs but not OHCs. However, exogenous Wnt4 treatment affects both IHCs and OHCs. How the loss of IHCs indirectly affects OHCs (lines 241-243)? In the discussion section, the authors need to further elaborate how Wnt4 interaction with Frizzled receptors inhibits both IHCs and OHCs.

We thank you reviewer for their comment. During hair cell development there is a medial to lateral gradient where IHCs are the first to develop followed by the rows 1-3 (in this order) of OHCs. In this manner, if IHCs do not develop it will lead to inhibition of OHC development. We have added the following to the manuscript for clarity starting at line 268 “Hair cell development occurs in a medial to lateral gradient where IHCs develop first followed by row1, row2, then row3 OHCs [1]. If IHCs do not develop this leads to inhibition of OHC development”.

We have also elaborated more on this point and added the following to the discussion (starting at line 614) “…it is likely that the greater degree of IHC inhibition also contributed to the secondary loss of OHCs where if IHCs do not develop this leads to a halt in the medial to lateral development of OHCs”.

Thank you for your suggestion. We have edited the manuscript to further elaborate on how Wnt4 could potentially interact with Frizzled receptors to inhibit both IHCs and OHCs starting on line 610 which reads:

In vitro, exogenous Wnt4 eliminated the in vivo Wnt4 gradient and inhibited the formation of both IHCs and OHCs. It is possible that exogenous Wnt4 can interact with Frizzled receptors in the lateral prosensory region through paracrine signaling and lead to inhibition of OHC differentiation.” Future studies can investigate this further.

  1. There is direct evidence that Wnt4 inhibits hair cell formation through regulation of calcium concentrations. Can calcium chelation rescue the loss of hair cells caused by exogenous Wnt4?

We thank the reviewer for the question. Calcium chelation using BAPTA-AM showed induction of ectopic inner hair cells as we have described in Figure 5. Additionally, PKC inhibition with BIM I was able to rescue the loss of HCs in explant cultures treated with Wnt4-CM (Figure 7).

Round 2

Reviewer 2 Report

Comments and Suggestions for Authors

The manuscript have revised according to the comments and maybe suitable for potential publication.

Reviewer 3 Report

Comments and Suggestions for Authors

I have no further comments.